# Association between use of antihypertensives and cognitive decline in the elderly—A retrospective observational study

**Prabhpaul Dhami[1], Kannayiram Alagiakrishnan[2]\*, Ambikaipakan Senthilselvan[3]**

**1** Department of Medicine, University of Alberta, Edmonton, Alberta, Canada, **2** Division of Geriatric Medicine, Department of Medicine, University of Alberta, Edmonton, Alberta, Canada, **3** School of Public Health, University of Alberta, Edmonton, Alberta, Canada

\* ka9@ualberta.ca

## Abstract

### Aim

Mild cognitive impairment (MCI) is the prodromal phase of dementia. The objective of this study was to determine whether specific antihypertensives were associated with conversion from MCI to dementia.

### Methods

In this retrospective study, a chart review was conducted on 335 older adults seen at the University of Alberta Hospital, Kaye Edmonton Seniors Clinic who were diagnosed with MCI. At the point of diagnosis, data was collected on demographic and lifestyle characteristics, measures of cognitive function, blood pressure measurements, use of antihypertensives, and other known or suspected risk factors for cognitive decline. Patients were followed for 5.5 years for dementia diagnoses. A logistic regression analysis was then conducted to determine the factors associated with conversion from MCI to dementia.

### Results

Mean age (± standard deviation) of the study participants was 76.5 ± 7.3 years. Patients who converted from MCI to dementia were significantly older and were more likely to have a family history of dementia. After controlling for potential confounders including age, sex, Mini Mental Status Exam scores and family history of dementia, patients who were on beta-blockers (BBs) had a 57% reduction in the odds of converting to dementia (OR: 0.43, 95% CI: 0.23, 0.81).

### Conclusions

In this study, BB use was protective against conversion from MCI to dementia. Further studies are required to confirm the findings of our study and to elucidate the effect of BBs on cognitive decline.

**Data Availability Statement:** Data were obtained from Alberta Health Services under strict guidelines for the use of the data for this study and authors are not permitted to share the data. Data requests

were obtained from research.administration@ahs.ca at Alberta Health Services after receiving the ethics approval from the University of Alberta Health Research Ethics Board (contact via reoffice@ualberta.ca).

**Funding:** Prabhpaul Dhami received the Summer Student Award from the Division of Geriatric Medicine, University of Alberta. The funders had no role in study design, data collection and analysis, decision to publish, or preparation of the manuscript.

**Competing interests:** The authors have declared that no competing interests exist.

## Introduction

Dementia is a disease that impacts people across the globe, and its prevalence is slowly increasing. The projected global prevalence of dementia is around 115 million in 2050 [1]. Mild cognitive impairment (MCI) is the prodromal phase of dementia [2]. Patients diagnosed with MCI, a stage between normal cognitive decline associated with aging and dementia, represents a high-risk group for the development of dementia. Up to 15% of patients with MCI convert to dementia yearly [3].

Hypertension, diabetes mellitus and obesity are strong risk factors for dementia. Both mid and late life high blood pressures (BPs) have been associated with dementia [4]. Additionally, low BP, and the type of BP lowering medications used may also impact dementia risk. Therefore, the risk of cognitive decline and conversion to dementia is not only determined by BP hemodynamics, but also by the type of antihypertensive medications used [5, 6]. In the SPRINT-MIND study conducted in the USA and Puerto Rico, intense systolic blood pressure (SBP) treatment targeted to less than 120 mmHg rather than 140 mmHg showed significantly lower incidence of cognitive decline [7].

BP parameters, along with other vascular risk factors, have been studied extensively in relation to dementia risk [8]. These modifiable risk factors have been the focus of primary and secondary prevention in patients presenting with cognitive complaints. As such, older adults are often prescribed antihypertensive medications if they present with hypertension. According to the literature, the potential differences of antihypertensives on cognitive function remains unclear. A systematic review has shown all antihypertensives reduce the incidence of dementia, with certain antihypertensive types, such as angiotensin receptor blockers (ARBs), being the most beneficial [9] Another systematic review points out, antihypertensives such as Calcium Channel Blockers (CCB), Beta-blockers (BB) and diuretics does not seem to improve cognitive function [10]. A study by Lee et al. showed that risk of dementia is higher in subjects who are on antihypertensives, however, the authors also mention this may be due to the confounding effect of comorbidities [11]. A case-control study indicated, some antihypertensives like ACE inhibitors and beta blockers may have a possible protective effect on the development of dementia. However, the relationship between type of antihypertensives with conversion from MCI to dementia largely remain unstudied [12]. This study will evaluate the effect of antihypertensives on conversion from MCI to dementia. We hypothesize that the type of antihypertensives may have an impact in the conversion from MCI to dementia.

## Materials and methods

This study is a retrospective chart review. Using the electronic medical record at the Kaye Edmonton Senior's Clinic, we retrieved 1330 patient charts from geriatric patient assessments conducted from 2015 to 2018. Inclusion criteria: patients who were diagnosed with MCI during baseline assessment were included in our study. Exclusion criteria: patients with dementia or those who did not have a cognitive assessment during these encounters were excluded from the study. We included 335 patients who were diagnosed with MCI during this baseline geriatric assessment. Data on antihypertensive use, BP readings, family history of dementia, known or suspected risk factors for cognitive decline, current medications, as well as demographic, lifestyle and social factors were also collected at baseline assessments. Patients were then followed up to 5.5 years to determine if they were diagnosed with dementia subsequently. Ethics approval was obtained from the ethics committee at the University of Alberta Hospital.

MCI and dementia diagnoses at baseline were made based on history, physical, neurological and cognitive testing, as well as brain imaging by geriatricians familiar with MCI and dementia diagnostic criteria [2, 13–16]. Diagnoses of dementia were done using DSM-IV or DSM-V

criteria, which includes: 1) short term memory deficits, 2) one or more of aphasia, apraxia, agnosia, and abstraction difficulties, 3) functional and social decline secondary to cognitive changes, and 4) no delirium or depression at the time of diagnosis (16). Diagnoses of MCI was done using the Petersen's or European Consortium Criteria, which includes: 1) cognitive complaints reported either by the patient or caregivers, 2) objective decline in memory or another cognitive domain as assessed by Montreal Cognitive Assessment (MoCA) and Mini Mental Status Exam (MMSE), 3) cognitive decline that does not impair daily life activities, and 4) the absence of dementia [2, 13–15]. BP readings were done by trained clinical nurses using digital BP machines in supine and standing positions after 5–10 minutes of rest in all subjects. Information about use of antihypertensive medications was obtained from the patients' charts.

## Statistical analysis

Baseline characteristics of patients in the study were described with mean and standard deviation values for continuous variables (e.g., BP parameters and age), proportions for binary categorical variables (e.g., antihypertensive use). Two-independent sample t-tests and chi-square tests were used to determine significant differences in continuous and categorical variables between those who converted from MCI to dementia and those who did not convert. A multivariable logistic regression analysis was conducted to determine whether specific antihypertensive medications were individually associated with conversion from MCI to dementia after controlling for potential confounders. Statistical analyses were conducted using STATA, and statistical significance was set at $p < 0.05$.

## Results

The mean (± standard deviation) age of the study participants was 76.5 ± 7.3 years with females making up a greater proportion of the subjects compared to males (54.6% vs. 45.4%). Among the participants, 30.2% had a family history of dementia. The type of dementia diagnosed among the 144 patients who converted from MCI to dementia included Alzheimer's Disease (AD) (38.9%), vascular dementia (11.1%), mixed dementia (22.9%), Lewy body dementia (3.5%), frontotemporal dementia (2.1%), Parkinsons disease dementia (2.1%), other (0.7%) and unknown (18.8%).

The distribution of demographic, lifestyle and other clinical characteristics for MCI converters and non-converters are illustrated in Table 1. Patients who converted from MCI to dementia were significantly older and more likely to have a family history of dementia. Of the antihypertensives evaluated in this study, only beta-blockers (BBs) showed a significant association with conversion from MCI to dementia. The proportion of patients who were on BBs was significantly lower among those converted from MCI to dementia relative to those who did not convert (13.9% vs. 25.7%, p = 0.008).

In addition, there was significant differences in age, sex, family history of dementia, MMSE scores, MoCA scores, SBP, and diastolic blood pressure (DBP) between patients who converted from MCI to dementia relative to non-converters. Significant factors identified in the univariate analysis were then included in the multiple logistic regression analysis. In the final logistic regression model, after adjusting for age, sex, family history of dementia and MMSE score, use of BBs remained significantly associated with conversion from MCI to dementia and shown in Table 2. Patients who were on BBs had a 57% reduction in the odds of conversion from MCI to dementia (OR: 0.43, 95% CI: 0.23, 0.81).

**Table 1. Baseline characteristics of MCI converter and MCI non-converter groups.**

| Variables | MCI Non-converters | MCI Converters | p-value |
|---|---|---|---|
| | (n = 191) | (n = 144) | |
| Demographic and Lifestyle Characteristics | | | |
| Age (mean ± SD) years | 75.7 ± 7.3 | 77.4 ± 7.1 | 0.04 |
| | (%) | (%) | |
| Sex | | | 0.34 |
| Female | 52.4 | 57.6 | |
| Male | 47.6 | 42.4 | |
| Current or past smoker | 39.3 | 45.8 | 0.23 |
| Family history of dementia | 21.1 | 42.6 | < 0.001 |
| Cognitive Assessment Scores | | | |
| MMSE (mean ± SD) | 26.9 ± 2.7 | 25.6 ± 3.6 | < 0.001 |
| MoCA (mean ± SD) | 22.3 ± 3.9 | 20.8 ± 4.3 | 0.001 |
| Vascular Risk Factors | | | |
| | (%) | (%) | |
| Overweight / Obese | 76.2 | 65.0 | 0.03 |
| Hypertension | 60.2 | 64.6 | 0.41 |
| Diabetes | 25.7 | 20.8 | 0.30 |
| Coronary Artery Disease | 17.3 | 14.6 | 0.51 |
| Cerebrovascular Disease | 13.1 | 13.2 | 0.98 |
| Orthostatic hypotension | 20.9 | 24.1 | 0.50 |
| Blood Pressure Parameters | | | |
| Supine Diastolic Pressure | | | 0.03 |
| (mean ± SD) | 73.2 ± 11.6 | 75.9 ± 10.3 | |
| Supine Systolic Pressure | | | <0.001 |
| (mean ± SD) | 131.8 ± 16.9 | 138.4 ± 17.4 | |
| Antihypertensive Use | | | |
| | (%) | (%) | |
| ACE Inhibitors | 28.8 | 29.2 | 0.94 |
| ARBs | 22.5 | 22.9 | 0.93 |
| CCBs | 18.3 | 25.7 | 0.10 |
| Alpha Blockers | 12.6 | 8.3 | 0.22 |
| Diuretics | 25.1 | 22.2 | 0.54 |
| Beta Blockers | 25.7 | 13.9 | 0.008 |

*ACE = angiotensin converting enzyme; ARB = angiotensin receptor blocker; CCB = calcium channel blockers

**Table 2. Risk factors for conversion of MCI to dementia: Results from the multivariable logistic regression analysis.**

| Variables | Odds ratio | 95% CI | p-value |
|---|---|---|---|
| Age | 1.03 | (1.00, 1.07) | 0.06 |
| Male sex | 0.91 | (0.56, 1.48) | 0.70 |
| Family History of Dementia | 3.42 | (2.02, 5.81) | <0.001 |
| MMSE | 0.87 | (0.80, 0.94) | 0.001 |
| Beta blockers | 0.43 | (0.23, 0.81) | 0.009 |

## Discussion

According to our study, patients with a higher SBP and DBP at baseline assessments were more likely to convert from MCI to dementia in the univariate analysis. However, this association was not seen after adjusting for other variables in the multivariate analysis. On the other hand, BB usage was significantly protective against conversion, as patients with MCI taking BBs at baseline were less likely to develop dementia in the follow up period.

Dementia and hypertension are common medical conditions that often co-exist in the elderly population. Long term literature evidence has shown that subjects with high blood pressure have a higher risk for cognitive decline and dementia [17]. As BP management is the standard practice for older adults with cognitive decline and dementia, the differential impact of each antihypertensive drug class on cognition is an important aspect to consider. However, the literature information on this topic has a lack of consistency [18], with many studies showing variable results regarding which antihypertensives are associated with dementia. The major types of antihypertensives studied on cognitive decline include diuretics, BB, CCB, angiotensin-converting enzyme inhibitors (ACEI) and ARB. Among these medications, CCBs and renin-angiotensin system (RAS) blockers/inhibitors have both been shown to prevent cognitive decline and dementia in a meta-analysis [18–20]. These drugs may play a role by specifically affecting AD pathology, as one study reported reduced amyloid plaque depositions on autopsy for patients treated with ARBs [21]. However, a recent ALLHAT trial follow-up study showed the risk of dementia did not vary significantly by using diuretics, CCBs, or ACEIs over a period of 18-years [22]. In our study, we did not see a significantly increased risk of conversion to dementia with any antihypertensive.

In fact, our study showed that BBs had a protective effect against conversion to dementia. The beneficial impact of BBs on cognitive impairment related to dementia is a topic that is relatively new in the literature. A 2013 study showed that hypertensive subjects treated with BBs had fewer Alzheimer's related brain lesions, reduced atrophy and less microinfarcts on autopsy compared to those using other antihypertensive medications [23]. A prospective cohort study has also demonstrated that BBs have a small protective effect against developing dementia [24]. In addition, there is evidence that certain BBs can be protective against AD by reducing amyloid plaque levels [25], one animal study confirmed this effect [25]. Contrary to the above findings, one animal study has pointed out that BBs can increase AD risk, which may be to the effect of beta-blockage reducing neurogenesis and worsening cognitive decline [26]. Another animal study points out that BBs have a complex immunomodulatory effect, with alteration of inflammatory signaling leading to progression of AD [27]. In searching for an explanation for these differential effects of medications on cognitive decline, it is important to consider the pharmacokinetic properties of the drug in question. For example, recent data has shown that those who specifically use blood-brain barrier permeable BBs for treatment of hypertension have a decreased incidence of subsequent AD [28]. This effect was specific for AD, and was not seen for other forms of dementia, which suggests that BBs impact the pathophysiology of the disease in some way. This theory has been corroborated by the literature as the production of amyloid plaques, a hallmark and possible cause of AD, is mediated by beta (2)-adrenergic receptor stimulation in the brain [29]. Thus, use of BBs, especially in MCI or early stages of AD, may slow down or prevent the accumulation of these amyloid plaques in the brain. While many antihypertensives have been noted to decrease the incidence of dementia in patients with vascular risk factors, BB in some preclinical and in few clinical studies shown the potential of reducing the prevalence of AD [30]. Although literature evidence on BBs and its effect on cognition is sparse, the potential cognitive benefits of using BBs in patients with MCI or dementia should be explored in future research studies.

## Limitations

One of the limitations in our report is that we have used a convenient sample because of the retrospective nature of the study. With the total sample size of 335 subjects, distributed between 191 patients in the non-converter group and 144 patients in the converter group, this study had an adequate statistical power of 80% or greater with a 5% level of significance. Thus, we were able to detect a difference of 12% or greater in the baseline characteristics between the two groups. Due to the small number of users of specific medications within each antihypertensive drug class, we could not examine the effect of specific medications on conversion from MCI to dementia. Additionally, with the retrospective design, we only measured outcomes in patients who were prescribed a hypertension medication at baseline, and subjects who were prescribed or discontinued medications during the follow up period would have been excluded.

## Conclusions

The mechanism by which hypertension impacts cognitive decline may be explained by specific BP parameters and types of antihypertensives used for treatment. This retrospective observational study has shown that BBs seem to decrease the risk of MCI conversion to dementia. The cognitive effects of antihypertensive medications need be considered in patients diagnosed with hypertension and comorbid cognitive impairment. This may include either initiating patients or switching to drugs that are associated with decreased incidence of dementia, such as BBs. Due to the widespread usage of antihypertensives, especially among the elderly, differential impacts of these medications on cognitive decline should be considered for all patients regardless of baseline risk. There is a need for future prospective studies and randomized controlled trials to confirm our findings and to further elucidate the effect of BBs on cognition.

## Author Contributions

**Conceptualization:** Kannayiram Alagiakrishnan.

**Data curation:** Prabhpaul Dhami, Ambikaipakan Senthilselvan.

**Formal analysis:** Ambikaipakan Senthilselvan.

**Funding acquisition:** Kannayiram Alagiakrishnan.

**Methodology:** Kannayiram Alagiakrishnan.

**Project administration:** Prabhpaul Dhami, Kannayiram Alagiakrishnan.

**Supervision:** Kannayiram Alagiakrishnan.

**Writing – original draft:** Prabhpaul Dhami, Kannayiram Alagiakrishnan, Ambikaipakan Senthilselvan.

**Writing – review & editing:** Prabhpaul Dhami, Kannayiram Alagiakrishnan, Ambikaipakan Senthilselvan.

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
