## [Decision Letter · Decision Letter 0]

26 Sep 2023

PONE-D-23-28542Association between use of antihypertensives and cognitive decline in the elderly -A retrospective observational studyPLOS ONE

Dear Dr. Alagiakrishnan,

Thank you for submitting your manuscript to PLOS ONE. After careful consideration, we feel that it has merit but does not fully meet PLOS ONE’s publication criteria as it currently stands. Therefore, we invite you to submit a revised version of the manuscript that addresses the points raised during the review process.

We look forward to receiving your revised manuscript.

Kind regards,

Masaki Mogi

Academic Editor

PLOS ONE

Journal Requirements:

2. We noticed you have some minor occurrence of overlapping text with the following previous publication(s) among others, which needs to be addressed:

ALAGIAKRISHNAN, K., DHAMI, P., SENTHILSELVAN, A.. Predictors of Conversion to Dementia in Patients With Mild Cognitive Impairment: The Role of Low Body Temperature. Journal of Clinical Medicine Research, North America, 15, apr. 2023. Available at: <https://www.jocmr.org/index.php/JOCMR/article/view/4883/25893713>. Date accessed: 14 Sep. 2023.

Cuevas C, Ryan N, Quirós A, Del Angel JG, Gonzalo N, Salinas P, Jiménez-Quevedo P, Nombela-Franco L, Nuñez-Gil I, Fernandez-Ortiz A, Macaya C, Escaned J. Determinants of percutaneous coronary intervention success in repeat chronic total occlusion procedures following an initial failed attempt. World J Cardiol 2017; 9(4): 355-362

Cuevas C, Ryan N, Quirós A, Del Angel JG, Gonzalo N, Salinas P, Jiménez-Quevedo P, Nombela-Franco L, Nuñez-Gil I, Fernandez-Ortiz A, Macaya C, Escaned J. Determinants of percutaneous coronary intervention success in repeat chronic total occlusion procedures following an initial failed attempt. World J Cardiol 2017; 9(4): 355-362 [PMID: 28515854 DOI: 10.4330/wjc.v9.i4.355]

In your revision ensure you cite all your sources (including your own works), and quote or rephrase any duplicated text outside the methods section. Further consideration is dependent on these concerns being addressed.

6. We note you have included a table to which you do not refer in the text of your manuscript. Please ensure that you refer to Table 2 in your text; if accepted, production will need this reference to link the reader to the Table.

Additional Editor Comments:

Two reviewers have an interest in this research; however, major concerns are raised in the present study. See the suggestions carefully, and respond to them appropriately.

Reviewers' comments:

Reviewer's Responses to Questions

**Comments to the Author**

1. Is the manuscript technically sound, and do the data support the conclusions?

Reviewer #1: Yes

Reviewer #2: No

2. Has the statistical analysis been performed appropriately and rigorously? 

Reviewer #1: Yes

Reviewer #2: No

3. Have the authors made all data underlying the findings in their manuscript fully available?

Reviewer #1: Yes

Reviewer #2: Yes

4. Is the manuscript presented in an intelligible fashion and written in standard English?

Reviewer #1: Yes

Reviewer #2: No

5. Review Comments to the Author

Reviewer #1: The authors examined the factors that contribute to the conversion from MCI to dementia and showed that the beta blocker is protective.

Ihave several major comments for this paper.

In line 68, please spell out CCBs.

In line 74, I think that 143 patients develop dementia after 5.5 years of observation of 335 MCI patients is too high a frequency. Please elaborate on the inclusion criteria for 335 patients.

In line 81, if the data for patients seen in 2021 were compiled in 2021, wouldn't it include patients with little or no observation period? The longer the observation period, the higher the risk of CONVERTER, so the observation period should be included in the COVARIATE.

In line 86, Please be specific about the diagnostic criteria, not just cite them, as the diagnosis of dementia and MCI is important.

In line 109 and Table 1, please describe what type of dementia the subject has developed, is it AD or DLB? Is it also vascular dementia?

In line 115, there was significant differences in age, sex, family history of dementia, MMSE scores, MoCA scores, SBP, and diastolic blood pressure (DBP) between patients who converted from MCI to dementia relative to non-converters. I think more attention should be paid to the higher blood pressure in the Converter group. Does Multiple logistic analysis include blood pressure?

In line 127, authors discussed that BB usage was significantly protective against conversion, as patients with MCI taking BBs at baseline were less likely to develop dementia in the follow up period. I think this is an oversimplification. Could it be considered that patients being treated with BB have a reduced risk of dementia due to better blood pressure control? It would be good to compare blood pressure for each patient on each drug to clarify this.

In line 146, a preclinical study showed that beta 2-adrenergic activation enhances neurogenesis (21). If this is true, wouldn't BB decrease neurogenesis and put us at risk for dementia?

In Discussion, please discuss the importance of blood pressure control as blood pressure is significantly higher in CONVERTER. Recently, it has been reported that hypertension may increase the risk of AD and other dementias by causing stagnation of the glymphatic pathway due to enlargement of the perivascular space (e.g. PMID: 37537887).

Reviewer #2: Thank you for allowing me to review the manuscript entitled “Association between use of antihypertensives and cognitive decline in the elderly -A retrospective observational　study.” This retrospective observational study revealed that the patients who were on beta-blockers (BBs) had a 57% reduction in the odds of converting to dementia. According to my opinion this study is unique, novel, and is of great interest. However, before this manuscript can be considered suitable for publication, several major concerns must be addressed.

1) Please revise the table to make it easier for readers to understand. What do the numbers in the table represent? I would like you to check the number and p-value in Current or past smokers because I think that they may be wrong.

2) I think that MCI is reversible. Therefore, the patients who have reverted to healthy individuals should be excluded. In addition, the patients whose type of antihypertensive drug has changed may be excluded too. The authors can identify them.

3) Did the authors include the all dementia? Why did not they just focus on AD?

4) I think it would be better to clearly state the hypothesis of this research in Background section.

5) There are differences in the backgrounds of the two groups. In my opinion, propensity score matching may be appropriate in this study. Please reconsider the statistical analysis.

6. PLOS authors have the option to publish the peer review history of their article (what does this mean?). If published, this will include your full peer review and any attached files.

Reviewer #1: No

Reviewer #2: No

---

## [Author Response · Author response to Decision Letter 0]

7 Nov 2023

Dear Editor,

 Thank you to the editor and reviewers for their suggestions. We have addressed all the suggestions and comments as described below, and all the changes in the manuscript are high lightened:

Journal Requirements:

Authors: We have made sure that the PLOS ONE’s style requirements have been incorporated in the manuscript. 

2. We noticed you have some minor occurrence of overlapping text with the following previous publication(s) among others, which needs to be addressed:

(a) ALAGIAKRISHNAN, K., DHAMI, P., SENTHILSELVAN, A.. Predictors of Conversion to Dementia in Patients With Mild Cognitive Impairment: The Role of Low Body Temperature. Journal of Clinical Medicine Research, North America, 15, apr. 2023. Available at: <https://www.jocmr.org/index.php/JOCMR/article/view/4883/25893713>. Date accessed: 14 Sep. 2023.

(b) Cuevas C, Ryan N, Quirós A, Del Angel JG, Gonzalo N, Salinas P, Jiménez-Quevedo P, Nombela-Franco L, Nuñez-Gil I, Fernandez-Ortiz A, Macaya C, Escaned J. Determinants of percutaneous coronary intervention success in repeat chronic total occlusion procedures following an initial failed attempt. World J Cardiol 2017; 9(4): 355-362

(c) Cuevas C, Ryan N, Quirós A, Del Angel JG, Gonzalo N, Salinas P, Jiménez-Quevedo P, Nombela-Franco L, Nuñez-Gil I, Fernandez-Ortiz A, Macaya C, Escaned J. Determinants of percutaneous coronary intervention success in repeat chronic total occlusion procedures following an initial failed attempt. World J Cardiol 2017; 9(4): 355-362 [PMID: 28515854 DOI: 10.4330/wjc.v9.i4.355]

In your revision ensure you cite all your sources (including your own works), and quote or rephrase any duplicated text outside the methods section. Further consideration is dependent on these concerns being addressed.

Authors: 

There is some overlap only in the methods section with the first article (Alagiakrishnan et al.). Our article does not have any overlap with article by Cuevas et al.

Authors:

We have removed funding information from the manuscript and included in the online submission form.

Authors:

Financial disclosure has been removed from the manuscript. Funding information and competing interest section have been updated in the cover letter.

Authors

According to restrictions placed on us by Alberta health services, we are not permitted to share any data regarding the patients in this study. The data we are allowed to make available is presented in the tables and manuscript. 

Data Availability statement: “The authors confirm that the data supporting the findings of this study are available within the article.”

6. We note you have included a table to which you do not refer in the text of your manuscript. Please ensure that you refer to Table 2 in your text; if accepted, production will need this reference to link the reader to the Table.

Authors:

We have now referred Table 2 in line 136 page 8 of the text.

Authors:

We do not have any supporting information files in the manuscript that require citations.

Additional Editor Comments:

Two reviewers have an interest in this research; however, major concerns are raised in the present study. See the suggestions carefully, and respond to them appropriately.

Reviewers' comments:

Reviewer's Responses to Questions

Comments to the Author

1. Is the manuscript technically sound, and do the data support the conclusions?

Reviewer #1: Yes

Reviewer #2: No

2. Has the statistical analysis been performed appropriately and rigorously?

Reviewer #1: Yes

Reviewer #2: No

3. Have the authors made all data underlying the findings in their manuscript fully available?

Reviewer #1: Yes

Reviewer #2: Yes

4. Is the manuscript presented in an intelligible fashion and written in standard English?

Reviewer #1: Yes

Reviewer #2: No

5. Review Comments to the Author

Please use the space provided to explain your answers to the questions above. You may also include additional comments for the author, including concerns about dual publication, research ethics, or publication ethics. (Please upload your review as an 

Reviewer #1: The authors examined the factors that contribute to the conversion from MCI to dementia and showed that the beta blocker is protective.

 Ihave several major comments for this paper.

In line 68, please spell out CCBs.

Authors:

We have now spelled out CCBs in line 65 in page 4 of the manuscript.

In line 74, I think that 143 patients develop dementia after 5.5 years of observation of 335 MCI patients is too high a frequency. Please elaborate on the inclusion criteria for 335 patients.

Authors:

We have now elaborated the inclusion/exclusion criteria in the methods section in lines 76 to 784, page 4 of our manuscript. 

In line 81, if the data for patients seen in 2021 were compiled in 2021, wouldn't it include patients with little or no observation period? The longer the observation period, the higher the risk of CONVERTER, so the observation period should be included in the COVARIATE.

Authors:

Patient charts from geriatric patient assessments conducted from 2015 to 2018 were retrieved for the study. The baseline of the study was between 2015 and 2018 and the patients were then followed up for at least three years with an average of 5 years. None of the patients seen for the first time in 2021 was included in the study.

In line 86, Please be specific about the diagnostic criteria, not just cite them, as the diagnosis of dementia and MCI is important.

Authors:

We have now included diagnostic criteria of MCI and dementia in the methods section, line 87 to 97, page 5 of our manuscript. 

In line 109 and Table 1, please describe what type of dementia the subject has developed, is it AD or DLB? Is it also vascular dementia?

Authors:

The types of dementia are now included in line 115-119, page 6, of the manuscript. 

In line 115, there was significant differences in age, sex, family history of dementia, MMSE scores, MoCA scores, SBP, and diastolic blood pressure (DBP) between patients who converted from MCI to dementia relative to non-converters. I think more attention should be paid to the higher blood pressure in the Converter group. Does Multiple logistic analysis include blood pressure?

Authors:

We initially included systolic and diastolic blood pressure in the logistic regression analysis. As they were no longer significant after accounting for other variables, they were removed from the final model.

In line 127, authors discussed that BB usage was significantly protective against conversion, as patients with MCI taking BBs at baseline were less likely to develop dementia in the follow up period. I think this is an oversimplification. Could it be considered that patients being treated with BB have a reduced risk of dementia due to better blood pressure control? It would be good to compare blood pressure for each patient on each drug to clarify this.

Authors:

In our study, systolic and diastolic blood pressure was included in the logistic regression, and they did not confound the association between use of beta-blocker and conversion from MCI to dementia. This indicates it is unlikely that better blood pressure control is why beta-blocker use reduced risk of dementia. 

In line 146, a preclinical study showed that beta 2-adrenergic activation enhances neurogenesis (21). If this is true, wouldn't BB decrease neurogenesis and put us at risk for dementia?

Authors:

Thank you for pointing out this information. We have clarified these points in the discussion section on line 166-175 page 10 of the manuscript. Although this one preclinical study points out that use of beta-blockers is associated with decreased neurogenesis and possible worsening of cognitive outcomes, a post-mortem study and a recent human study points out the possibility of a protective effect, which is also seen in our study. We have added these references to the manuscript as well.

In Discussion, please discuss the importance of blood pressure control as blood pressure is significantly higher in CONVERTER. Recently, it has been reported that hypertension may increase the risk of AD and other dementias by causing stagnation of the glymphatic pathway due to enlargement of the perivascular space (e.g. PMID: 37537887).

Authors:

We have now elaborated on the importance of BP control, and included an additional reference in line 186-187 in page 10,11 of our manuscript.

Reviewer #2: Thank you for allowing me to review the manuscript entitled “Association between use of antihypertensives and cognitive decline in the elderly -A retrospective observational　study.” This retrospective observational study revealed that the patients who were on beta-blockers (BBs) had a 57% reduction in the odds of converting to dementia. According to my opinion this study is unique, novel, and is of great interest. However, before this manuscript can be considered suitable for publication, several major concerns must be addressed.

1) Please revise the table to make it easier for readers to understand. What do the numbers in the table represent? I would like you to check the number and p-value in Current or past smokers because I think that they may be wrong.

Authors

We have revised Table 1 to make it easier for the readers to understand. We thank the reviewer for alerting us to the incorrect frequencies and p-values for current and past smokers. We have now corrected the error. This change can be found on page 7 of the manuscript.

2) I think that MCI is reversible. Therefore, the patients who have reverted to healthy individuals should be excluded. In addition, the patients whose type of antihypertensive drug has changed may be excluded too. The authors can identify them.

Authors:

In our study, none of the patients reverted to normal cognitive function for age. Patients were diagnosed at the Seniors Clinic based on subjective/objective criteria (as described in text) by trained geriatricians familiar with MCI diagnosis. 

3) Did the authors include the all dementia? Why did not they just focus on AD?

Authors:

We included all types of dementia in line 115-119 on page 6 of the manuscript. The sample size was not adequate to focus on one type of dementia.

4) I think it would be better to clearly state the hypothesis of this research in Background section.

Authors:

We have now added the hypothesis information at the end of the introduction section of the article in line 72-73 on page 4.

5) There are differences in the backgrounds of the two groups. In my opinion, propensity score matching may be appropriate in this study. Please reconsider the statistical analysis.

Authors:

The sample size is too small to conduct propensity score analysis. There were only 69 beta-blockers users, and this sample size is too small to consider beta-blockers as the main exposure in the propensity score analysis.

---

## [Decision Letter · Decision Letter 1]

14 Nov 2023

PONE-D-23-28542R1Association between use of antihypertensives and cognitive decline in the elderly -A retrospective observational studyPLOS ONE

Dear Dr. Alagiakrishnan,

Thank you for submitting your manuscript to PLOS ONE. After careful consideration, we feel that it has merit but does not fully meet PLOS ONE’s publication criteria as it currently stands. Therefore, we invite you to submit a revised version of the manuscript that addresses the points raised during the review process.

Major revisions according to the Reviewer's comments are still necessarfy in the present form.See the comments and respond them appropriately.

We look forward to receiving your revised manuscript.

Kind regards,

Masaki Mogi

Academic Editor

PLOS ONE

Reviewers' comments:

Reviewer's Responses to Questions

**Comments to the Author**

1. If the authors have adequately addressed your comments raised in a previous round of review and you feel that this manuscript is now acceptable for publication, you may indicate that here to bypass the “Comments to the Author” section, enter your conflict of interest statement in the “Confidential to Editor” section, and submit your "Accept" recommendation.

Reviewer #2: All comments have been addressed

2. Is the manuscript technically sound, and do the data support the conclusions?

Reviewer #2: No

3. Has the statistical analysis been performed appropriately and rigorously? 

Reviewer #2: No

4. Have the authors made all data underlying the findings in their manuscript fully available?

Reviewer #2: Yes

5. Is the manuscript presented in an intelligible fashion and written in standard English?

Reviewer #2: Yes

6. Review Comments to the Author

Reviewer #2: Thank you for allowing me to review the revised manuscript entitled “Association between use of antihypertensives and cognitive decline in the elderly -A retrospective observational study.” I think that the authors generally responded to comments. In this study, the small sample size might cause the statistical analysis problem, which may change the interpretation of the results. Although the authors mention the sample size in the limitation section, it's not enough and I would like you to discuss fully.

7. PLOS authors have the option to publish the peer review history of their article (what does this mean?). If published, this will include your full peer review and any attached files.

Reviewer #2: No

---

## [Author Response · Author response to Decision Letter 1]

22 Nov 2023

Review Comments to the Author

Reviewer #2: Thank you for allowing me to review the revised manuscript entitled “Association between use of antihypertensives and cognitive decline in the elderly -A retrospective observational study.” I think that the authors generally responded to comments. In this study, the small sample size might cause the statistical analysis problem, which may change the interpretation of the results. Although the authors mention the sample size in the limitation section, it's not enough and I would like you to discuss fully.

Authors:

We thank the reviewer for the positive comment about our response to the reviewer’s previous comments. We have now included the following statement in the limitation section on the statistical power of our study with a sample size of 335 subjects. 

“In this study, the total sample size of 335 subjects with 191 in the non-converter group and 144 in the converter group had an adequate statistical power of 80% or greater with 5% level of significance to detect a difference of 12% or greater in the baseline characteristics between the two groups. Due to the small number of users of specific medications within each antihypertensive drug class in this study, we could not examine the effect of specific medications on conversion from MCI to dementia

---

## [Decision Letter · Decision Letter 2]

24 Nov 2023

Association between use of antihypertensives and cognitive decline in the elderly -A retrospective observational study

PONE-D-23-28542R2

Dear Dr. Alagiakrishnan,

We’re pleased to inform you that your manuscript has been judged scientifically suitable for publication and will be formally accepted for publication once it meets all outstanding technical requirements.

Kind regards,

Masaki Mogi

Academic Editor

PLOS ONE

Additional Editor Comments (optional):

No further comment.

Reviewers' comments:

Reviewer's Responses to Questions

**Comments to the Author**

1. If the authors have adequately addressed your comments raised in a previous round of review and you feel that this manuscript is now acceptable for publication, you may indicate that here to bypass the “Comments to the Author” section, enter your conflict of interest statement in the “Confidential to Editor” section, and submit your "Accept" recommendation.

Reviewer #2: All comments have been addressed

2. Is the manuscript technically sound, and do the data support the conclusions?

Reviewer #2: Yes

3. Has the statistical analysis been performed appropriately and rigorously? 

Reviewer #2: Yes

4. Have the authors made all data underlying the findings in their manuscript fully available?

Reviewer #2: Yes

5. Is the manuscript presented in an intelligible fashion and written in standard English?

Reviewer #2: Yes

6. Review Comments to the Author

Reviewer #2: Thank you for your appropriate response. I think that the revised manuscript is improved. I have no more comments.

7. PLOS authors have the option to publish the peer review history of their article (what does this mean?). If published, this will include your full peer review and any attached files.

Reviewer #2: No

---

## [Editor Report · Acceptance letter]

11 Dec 2023

PONE-D-23-28542R2 

Association between use of Antihypertensives and Cognitive Decline in the Elderly - A Retrospective Observational Study 

Dear Dr. Alagiakrishnan:

I'm pleased to inform you that your manuscript has been deemed suitable for publication in PLOS ONE. Congratulations! Your manuscript is now with our production department. 

Kind regards, 

on behalf of

Dr. Masaki Mogi 

Academic Editor

PLOS ONE